# Gastrointestinal Cancer Patient Nutritional Management: From Specific Needs to Novel Epigenetic Dietary Approaches

**DOI:** 10.3390/nu14081542

**Published:** 2022-04-08

**Authors:** Chiara Cencioni, Ilaria Trestini, Geny Piro, Emilio Bria, Giampaolo Tortora, Carmine Carbone, Francesco Spallotta

**Affiliations:** 1Institute for Systems Analysis and Computer Science “A. Ruberti”, National Research Council (IASI-CNR), 00185 Rome, Italy; 2Section of Oncology, Department of Medicine, University of Verona Hospital Trust, 37134 Verona, Italy; ilaria.trestini@aovr.veneto.it; 3Medical Oncology, Department of Medical and Surgical Sciences, Fondazione Policlinico Universitario Agostino Gemelli IRCCS, 00168 Rome, Italy; geny.piro@policlinicogemelli.it (G.P.); emilio.bria@policlinicogemelli.it (E.B.); giampaolo.tortora@policlinicogemelli.it (G.T.); carmine.carbone@policlinicogemelli.it (C.C.); 4Medical Oncology, Department of Translational Medicine, Catholic University of the Sacred Heart, 00168 Rome, Italy

**Keywords:** gastrointestinal cancers, pancreatic cancer, diet, epigenetics

## Abstract

Nutritional habits impinge on the health of the gastrointestinal (GI) tract, contributing to GI disorder progression. GI cancer is a widespread and aggressive tumor sensitive to nutritional changes. Indeed, specific nutritional expedients can be adopted to prevent GI cancer onset and to slow down disease activity. Moreover, the patient’s nutritional status impacts prognosis, quality of life, and chemotherapy tolerance. These patients encounter the highest frequency of malnourishment risk, a condition that can progressively evolve into cachexia. Clinical studies dealing with this topic stressed the importance of nutritional counseling and put under the spotlight nutrient delivery, the type of nutrient supplementation, and timing for the start of nutritional management. A medical practitioner well-prepared on the topic of nutrition and cancer should operate in the clinical team dedicated to these oncological patients. This specific expertise needs to be implemented as soon as possible to adopt nutritional interventions and establish a proper patient-tailored dietary regimen. The nutritional gap closure should be prompt during anticancer treatment to stabilize weight loss, improve treatment tolerability, and ameliorate survival rate. Recently, novel nutritional approaches were investigated to target the bidirectional link between epigenetics and metabolism, whose alteration supports the onset, progression, and therapeutic response of GI cancer patients.

## 1. Introduction

The digestive system is responsible for processing food through absorption. Therefore, gastrointestinal tracts and diet composition establish complex interactions daily. Their crosstalk might influence gastrointestinal tract health. Indeed, those who suffer from gastrointestinal disorders perceive the diet as being responsible for symptoms and look for a nutritionist who can help to establish a suitable dietary regimen relieving gastrointestinal problems [1]. Unfortunately, it is difficult to find medical practitioners well prepared on the topic of diet and nutrition. Noteworthily, dietary factors in early life might represent risk factors of later health or disease [2,3]. For example, food quality and nutritional habits adopted in childhood or adolescence might affect the later development of an inflammatory bowel. Moreover, alimentary behaviors affect the symptoms and progression of gastrointestinal disorders, representing a target to slow down disease activity [2,3].

The dual role of nutrition, as a determinant and as a support in patient management, is of particular interest for gastrointestinal (GI) cancers, a collective term used for cancers affecting different gastrointestinal tracts, including the colorectum, small intestine, stomach, liver, esophagus, and pancreas [4,5]. GI cancers show higher mortality than any other kind of cancer. In 2020, they accounted for an estimated 3.5 million deaths worldwide, with a further 5.0 million new cases diagnosed in the same year [6]. Colorectal cancer (CRC) is the most common type of GI cancer, being the third most common of all organ cancers after lung and breast cancers, whereas gastric, liver, esophageal, and pancreatic cancers are ranked the fifth, sixth, eighth, and 12th most commonly diagnosed cancers, respectively [6]. Interestingly, nutritional habits and the body mass index value strongly correlate with GI cancer incidence and represent exogenous modifiable key factors to take under consideration, not only for primary prevention, but also as support in therapeutic interventions [4,5]. GI cancer patients experience a loss of appetite, early satiety, mouth sores, changes in taste and smell, epigastric pain, nausea, vomiting, constipation, and diarrhea [7,8]. Taken together, these are collectively indicated as nutrition impact symptoms (NIS) [7,8]. NIS associate with early stages of malnutrition and should be taken under control as GI cancer prognostic factors of the nutritional status [7,8].

The relationship between GI cancer and diet has been extensively studied, especially for the role of diet composition as a risk factor for these particular oncological patients [4,5]. More recently, several studies have addressed the effect of nutritional methods as therapeutic tools able to prevent disease progression and/or reduce side effects of chemotherapy [4,5]. The present review aims to point out the potential of these measures, highlighting the nutrient support needs of GI cancer patients and what is emerging from the already ongoing clinical trials on the topic. Moreover, a specific section will revise literature dealing with clinical and preclinical studies concerning nutritional therapy approaches aimed at targeting epigenetic and metabolic alterations observed in GI cancer patients.

## 2. Diet Intervention in GI Cancer Therapy

According to clinical data, GI cancer patients experience weight loss and signs of malnutrition at hospital admission, since nutrient digestion and absorption do not work properly [4,5,9]. Indeed, the entity of undernourishment strongly correlates with poor prognosis; higher frequency of postoperative complications; and increase in morbidity, mortality, and length of hospital stay [5,10,11]. Malnutrition is caused by a deficiency of essential nutrients, leading to a lack of sufficient protein intake essential for energy balance and to avoid muscle tone deprivation. Most cancer patients develop NIS as a sign of tumor presence/staging and of treatment received, which compromise oral intake [8]. Moreover, malnutrition affects mesenteric membrane function; pharmacodynamics; immune defenses, and physiology of the liver, the kidney, and the heart [12,13,14]. Often, the outcome is cachexia, a complex syndrome characterized by a heavy condition of debilitation due to weight loss associated with a loss of skeletal muscle mass. Two key mechanisms might contribute to cachexia: the breakdown of molecules by catabolism and inflammation, controlled by the immune system [15]. Catabolism results from a combined action of factors derived from either the tumor or the host in response to reduced nutrient availability, anti-neoplastic treatment, and physical activity. The complex interplay among these multiple signaling pathways alters the physiological balance between protein synthesis and breakdown rates, leading to onset of a protein hypercatabolic state [15]. Moreover, the pathogenesis of cancer cachexia is influenced by adipose tissue. Indeed, brown adipose tissue contains lipid droplets able to generate energy from proton leakage pathway initiation in the mitochondrial membrane [16]. This process induces reactive oxygen species’ release into the skeletal muscle, adipose tissue loss, and cancer energy imbalance. Furthermore, the reduced nutrient intake leads to increased catabolism of the stored fat, thus contributing to white adipose tissue as well as muscle mass loss [17].

Indeed, Fearon et al. described cachexia as a multifactorial syndrome defined by an ongoing loss of skeletal muscle mass, with or without a loss of fat mass, which cannot be completely reversed by conventional nutritional support and which leads to progressive functional impairment [9]. It might evolve, and it has been proposed that one can differentiate early phases without discernible weight loss (pre-cachexia) from advanced or refractory stages [18]. The agreed diagnostic criterion for cachexia includes weight loss greater than 5% of the usual body weight or weight loss greater than 2% in individuals already showing depletion according to current body weight and height (body–mass index (BMI) < 20 kg/m^2^) [9]. Cachexia leads to mortality increase and a poor quality of life. Of note, GI cancer and, in particular, pancreatic cancer patients encounter the highest frequency of developing cachexia.

In this light, careful management of nutrition is specifically relevant for these oncological patients to improve their prognosis, quality of life, and tolerance to anticancer treatments. First of all, these cancer patients should be screened to assess their nutritional risk. Several tools for nutritional screening have been developed, including the Malnutrition Universal Screening Tool (MUST), Nutrition Risk Screening 2002 (NRS-2002), and the Malnutrition Screening Tool (MST), but there is no general agreement on the ‘best’ screening tool [19,20]. In 2019, the Global Leadership Initiative on Malnutrition (GLIM) criteria for the diagnosis and grading of malnutrition were introduced. A patient is defined as malnourished if, after a positive risk-screening test, they present at least one phenotypic criterion (non-volitional weight loss (WL), low body mass index (BMI), or reduced muscle mass) and one etiologic criterion (reduced food intake/assimilation or inflammation/disease burden) [13]. All patients diagnosed as being at risk of malnutrition must be submitted to an in-depth nutrition assessment, including nutrient balance, body composition with a focus on muscle mass, and information regarding the presence and degree of systemic inflammation [21]. Moreover, an assessment of factors that are impeding or that might interfere with maintaining nutritional status should include an evaluation of NIS (such as anorexia, nausea, taste and smell alterations, and mucositis) as well as a clinical evaluation to assess maldigestion-related symptoms and nutritional deficiencies [22].

The revision of recent clinical trials on nutrition in patients with GI cancer showed studies mainly addressing nutrient delivery, type of nutrient supplementation, timing for the start of nutritional management, and the combination of specific nutrients with muscular exercise (Table 1). Different clinical trials deal with types of feeding comparing oral, enteral, and parental nourishing with the goal to minimize body weight loss; improve blood test results and chemotherapy compliance; increase quality of life; and decrease infection frequencies, hospitalization, and mortality (NCT02066363; NCT03949907; NCT04607057; NCT03150615; NCT01222208; NCT00919659; NCT01870817; NCT02155140) [23,24,25,26,27,28]. Temporarily beneficial effects have been observed in the case of parental feeding on nutritional status stabilization and quality of life [23,27]. Jejunostomy feeding contributes to satisfy GI cancer patient nutritional needs without compromising oral nutritional intake [25]. It has also been observed that after pancreaticoduodenectomy, early oral feeding might be feasible and safe, being able to decrease the prevalent complication of this surgery procedure represented by delayed gastric emptying.

Among nutritional approaches, there are clinical trials dealing with the supplementation of particular nutrients into the daily diet of GI cancer patients (NCT03863236; NCT04218253; NCT03930888; NCT02681601; NCT0490121; NCT01218841; NCT04732442; NCT01830907; NCT04513418; NCT02626195; NCT04567459; NCT04131426) [24,29,30]. These specific nutrients are represented by active agents, which might counteract the side effects of GI cancer and improve the quality of life of these patients. The calibration of the appropriate protein amount intake is crucial to preserve muscular tone, together with a specific schedule of a muscle exercise program (NCT04190121; NCT02788955; NCT04131426; NCT03475966; NCT05030090; NCT01276795). Furthermore, the right vitamin and anti-oxidant agent supply might be of particular relevance for the diet of GI cancer patients. Specifically, diet is supplemented with natural vitamin E (d-tocopherol triglyceride) or vitamin D and omega-rich food containing a high concentration and high purity of natural omega-3 polyunsaturated fatty acids, like eicosapentaenoic acid and docosahexaenoic acid [10,11,31]. All these substances show anti-inflammatory effects, which can preserve some defense capabilities against infections and decrease the postoperative release of inflammatory molecules, ultimately shortening the length of hospital and intensive care unit stay, as well as infectious complications [10,11,31]. These goals are pursued by also applying the so-called immunonutrition based on the supplementation of specific immunomodulatory molecules, which might enhance the release of active cytokines and the homing of leukocytes into the tumor mass (NCT01704664; NCT04732442; NCT04513418; NCT03550482) [32,33].

The number of clinical trials dealing with the topic of nutrition in GI cancer patients indicated the importance of assessing their risk of malnourishment and promptly adopting a specific nutritional strategy in the presence of NIS.

## 3. Diet as an Intervention to Improve Quality of Life of the Pancreatic Cancer Patient

As stated above, nutritional derangements are common presentation hallmarks of GI cancer. Upper GI cancers pose the highest risk to the development of malnutrition, and studies have shown that 22% of patients are severely malnourished and 63% moderately malnourished or at risk of malnutrition. Lower GI cancers pose a lower risk, with 10–17% being severely malnourished and 25–60% being moderately malnourished or at risk of malnutrition [34]. Nutritional wasting is particularly exacerbated in pancreatic cancer (PC). Indeed, unintentional weight loss with depletion of fat stores and muscle mass develops in over 80% of PC patients and might be the first sign leading to the diagnosis of this malignancy [35]. Recent advancements in image-based technologies and body composition research have facilitated the understanding of the high prevalence of cancer cachexia in PC patients, with an overall occurrence between 19% and 68% in both resectable and nonresectable patients and an early onset before the occurrence of a clinically apparent weight loss [14,35,36]. The identification of muscle wasting might be masked, because 40–60% of PC patients are overweight or obese, even in the metastatic setting [37]. Of note, sarcopenic obesity, defined as a loss of muscle mass and/or decline of muscle strength coupled with high levels of adiposity (obesity), negatively associates with both surgical complications and chemotherapy toxicities [38,39,40].

Among GI patients, several pathophysiological derangements might result in nutritional deterioration, and several factors often occur at the same time, including impaired food intake, due to centrally mediated pathways leading to anorexia and mechanical factors (esophageal and gastric cancer influence the ability to eat and drink); reduction in daily physical activity and its associated anabolic effects; as well as metabolic changes leading to systemic inflammation and activation of catabolism [9]. Moreover, some types of GI cancer impact nutrient digestion and absorption. The absorption of iron might be reduced when gastric acid is suppressed following partial gastrectomy [41]. Vitamin B_12_ absorption is affected by the loss of intrinsic factors due to the presence of disease or gastric surgery [42]. Pancreatic exocrine insufficiency (PEI) represents an additional cause of malnutrition in PC patients, occurring when the exocrine pancreas is unable to maintain its normal digestive function. It is a multifactor condition, involving loss of pancreatic parenchyma and/or obstruction of the main pancreatic duct (which impede either the production of pancreatic enzymes or their transfer into the duodenum), decreased pancreatic stimulation, or acid-mediated inactivation of pancreatic enzymes [43]. Untreated PEI results in maldigestion and malabsorption of nutrients, which manifest as abdominal bloating or discomfort and changes in bowel movements [22]. Of interest, PEI is a critical host factor in determining the intestinal microbiota composition, which can modulate tumor sensitivity to therapeutic agents [44,45].

The removal of any part of the small intestine or formation of a stoma influences the ability to absorb macronutrients, vitamins (such as vitamin B_12_, which is absorbed from the terminal ileum), electrolytes, fluids, and trace elements [46].

### 3.1. Malnutrition Status Impacting GI Cancer Patient Prognosis

As already mentioned, nutritional worsening not only represents symptoms of the disease but also is an important factor having a significant impact on the outcome of GI cancer patients and, in particular, PC patients [47]. An impaired nutritional status is strongly associated with a reduced ability to receive medical treatments and an increased risk of postsurgical complications with prolonged hospitalization, reduced performance status, and poor quality of life [35,48,49]. Specifically, low muscle mass contributes to poor surgical outcomes [50], a low tolerance to adjuvant chemotherapy [51], and earlier recurrence of disease [52]. Moreover, it is a relevant prognostic factor for overall survival (OS) in PC patients treated with both gemcitabine-based and FOLFIRINOX-like (leucovorin, fluorouracil, irinotecan, and oxaliplatin) chemotherapies [53,54,55]. Intriguingly, decreased skeletal muscle mass is believed to influence the host immune system, leading to immune senescence [56]. Of note, preclinical findings have highlighted the essential role of amino acids such as serine or L-arginine in promoting T-cell expansion and enhancing CD8þ T-cell antitumor activity [57,58]. Considering this, it is important to detect the risk of malnutrition and, eventually, to assess its presence by routinely implementing a standardized screening procedure in patients with PC to identify those who are either malnourished or at significant risk for malnourishment, since early stages (pre-cachexia) are probably the most susceptible to a nutritional therapeutic approach [59].

### 3.2. Nutritional Interventions

Given the complex and multifaceted contributors to nutritional derangements in GI, treatment requires a multitargeted and multidisciplinary approach with the aim of ensuring adequate energy and nutrient intake, minimizing catabolic alterations, and preventing or treating muscle wasting [60]. Nutritional intervention should be administered earlier in the disease trajectory when the window of anabolic potential is open [61]. Unfortunately, in clinical practice, many patients are evaluated too late, when several nutritional impairments have already been reported and the refractory stage of cancer cachexia has already occurred, rendering available interventions ineffective [62]. In this context, emerging evidence, from pre-clinical and clinical data, shows that early closing of the nutritional gap during anticancer treatment can stabilize weight loss, improve treatment tolerability, reduce the performance status deterioration, and ameliorate the survival rate [59,63].

Even though inadequate nutritional intake is not the only contributor to nutritional depletion in cancer patients, the optimal intake is essential for optimizing the nutritional intervention [12]. The caloric requirement of GI patients should be assessed in a personalized way, and when energy expenditure is not individually measured, the European Society for Clinical Nutrition and Metabolism (ESPEN) guidelines suggest an intake of 25–30 kcal/kg/day [64]. Davidson et al. reported that unresectable PC patients with stable weight showed significantly higher energy intakes (29.9 kcal/kg/day vs. 25.6 kcal/kg/day) and better survival outcomes [65]. Adequate energy intake is needed not only to avoid weight loss, but also to maintain muscle mass by stimulating protein synthesis and suppressing protein breakdown [61]. Data on nutritional support in oncological patients attribute high relevance to the correct protein intake to promote muscle protein synthesis. ESPEN guidelines on nutrition in cancer patients set the protein intake target to 1.2–1.5 g/kg per day [64]. However, the optimal protein intake for preventing or treating muscle depletion in cancer has not been determined. In this regard, a recent systematic review, including patients with cancer types with a high prevalence of muscle loss, showed that a protein intake over a cut-off between 1.2 and 1.4 g/kg per day might be necessary to avoid muscle wasting during treatment [12]. Moreover, a recent prospective cohort study in 38 patients with unresectable PC receiving chemotherapy revealed that an insufficient protein intake (<1.1 g/kg per day) was identified as an independent poor prognostic factor [66]. Promising research has highlighted the potential role of branched-chain amino acids (BCAAs) to decrease muscle catabolism [67,68], as well as β-hydroxy-β-methylbutyrate (HMB), a metabolite of the essential amino acid leucine, and its anabolic effect in cancer patients [69,70,71]. Nevertheless, to date, it has not been clarified if there are specific amino acid mixtures that can improve clinical outcomes in this setting [64]. High-quality research is needed to clarify an optimal dose–response and to understand the particular amino acids’ anabolic effect on GI patient outcomes.

Patient-tailored dietary counseling by a registered dietician should be the first choice of nutritional intervention offered to improve oral intake in patients who can eat. Dietary counseling should emphasize protein intake and an increased number of meals per day, manage NIS, and offer oral nutritional supplements in patients who cannot meet their nutritional requirements despite dietary intervention [18].

Among PC patients, early dietary counseling with oral nutritional supplements (ONS), when necessary, might improve not only overall macronutrient intake, nutritional status, and body composition, as suggested by a recent review by Kasvis et al., but also survival [72]. However, most trials on this topic were hampered by poor methodological quality, specifically from inadequate reporting of the actual dietary intake and not reaching recommended dietary intakes. Omega-3 fatty acids from fish oil have been studied, particularly for their anti-inflammatory properties, and they are available as components of ONS, usually enriched in protein. Particularly, eicosapentaenoic acid (EPA) has been shown to block ubiquitin–proteasome-induced muscle proteolysis, which could be suggestive of its favorable impact on muscle conservation in wasting syndromes [11]. Most studies have suggested that these ONS might aid in weight and lean body mass stabilization or gain [10,31]. Overall, trials were heterogeneous and inadequately powered to show effects on treatment toxicity or survival. Further research is required to elucidate the benefit of omega-3, independent of increased protein consumption [18].

Furthermore, the use of immunonutrition and its potential role in the modulation of inflammation and immunosuppression at the tumor microenvironment level in cancer patients has been progressively gaining attention as a high-calorie–high-protein nutritional blend enriched in immunonutrients. The most important are omega-3 fatty acids, arginine, and nucleotides. In the clinical setting of PC patients undergoing surgery, a recent systematic review and meta-analysis of randomized controlled trials revealed that immunonutrition significantly decreased the rate of infectious complications (RR = 0.47, 95% CI (0.23, 0.94), *p* = 0.03) and the length of hospital stay (MD = −1.90, 95% CI (−3.78, −0.02), *p* = 0.05), modulating the immune system, especially in the preoperative period [73]. However, well-designed randomized control trials are required to clarify the effect of immunonutrition in PC. Further, in patients who are unable to tolerate sufficient oral food intake, prompt artificial nutrition is recommended. Enteral nutrition should be preferable over parenteral nutrition due to lower costs and potential complications. Parenteral nutrition should be considered if the oral intake and tube feeding are not tolerated or remain inadequate [74]. To our knowledge, the impact of overnight, home-based parenteral nutrition on nutritional status in PC patients was reported in two studies [27,75]. Both studies reported weight maintenance or gain in most of the patients, although Richter et al. observed these positive results in only those with a survival of more than 5 months [75].

### 3.3. Pancreatic Enzyme Replacement Therapy

In addition to the nutritional interventions, pancreatic enzyme replacement therapy (PERT) might be required to relieve maldigestion-related symptoms and to preserve and/or improve nutritional status. Unfortunately, PERT is largely underused: a retrospective analysis in the metastatic PC setting showed very low rates of PERT prescription (21%), even though most patients had tumors of the pancreatic head and were likely to have an obstructed pancreatic duct [76]. Moreover, even when PERT is prescribed in patients presenting PEI, its dosage is sub-optimal in more than half of the patients [77]. A panel of experts in this field recently suggested that empiric treatment with PERT should be started, without prior testing, in all patients with a pancreatic head resection or tumor in the head of the pancreas and when there is a clinical suspicion of PEI, based on the typical symptoms and signs of malabsorption and malnutrition. Besides, a diagnostic evaluation should be performed using fecal elastase in patients affected by a body or tail neoplasm before giving them PERT [43]. Capsules should be swallowed during meals, rather than before or after the meal. PERT should start with doses of 40,000–50,000 units of lipase with meals and 10,000–25,000 units with every snack [43]. The dosage needs to be carefully monitored, as well as altered, depending on the patient food intake/pattern of eating, method of cooking, and portion sizes. This will require repeated educational visits regarding alteration of the dosage and timing of administration [78]. Clinical evidence suggests that PERT might prevent weight loss, induce weight gain, and is associated with an improved quality of life and, possibly, survival among patients with PC [22,79,80,81,82].

All these pieces of evidence suggest that attention to the nutritional status of these oncological patients is crucial to improve their quality of life as well as tolerance to anti-neoplastic drugs.

## 4. Epi-Metabolic Diet Approaches in GI Cancer Patients

An emerging field in the context of diet as a support for oncological patients is represented by novel approaches aimed to modulate the epigenetic and metabolic status, whose alteration contributes to the onset, progression, and therapeutic response of GI cancer. Indeed, metabolic as well as epigenetic reprogramming is a typical feature of cancer cells in general [83,84,85]. Metabolism and epigenetics are strictly interconnected and give rise to an intense bidirectional crosstalk by which they control each other [86]. Specifically, metabolic reprogramming is sustained by epigenetic derangement, since the alteration of epigenetic enzyme functions might affect the expression and activity of metabolic enzymes by chromatin remodeling and post-translational modification. Their interplay might lead to an unbalanced ratio of metabolites favoring an enhancement of oncometabolite levels. On the other hand, the presence of specific metabolites might affect the tumor epigenome, since most epigenetic enzymes exploit metabolites as co-factors or substrates to carry out their function. This epigenetic–metabolic (epi-metabolic) link indicates the strict relationship among nutrition, metabolism, and gene regulation to be taken into consideration in GI cancer research studies. In this light, dietary interventions affecting the epi-metabolic link might support the management of GI cancer patients, modifying progression and therapeutic responses as well as their quality of life.

The availability of nutrients containing acetate, butyrate, curcumin, NAD+, methionine, folate, ascorbate, and 2-oxoglutarate impacts the epi-metabolome. Their intake can be modulated in the dietary composition potentially supporting the targeting of specific GI cancer vulnerabilities in a combination with conventional anti-neoplastic therapies to enhance their cytotoxicity potential. Acetate is a precursor of acetyl-CoA, which is essential for histone acetyltransferase activity. Curcumin and butyrate are known histone deacetylase inhibitors with recognized anti-cancer properties. NAD+ supports the function of sirtuins, which are class III histone deacetylases. Methionine and folate are precursors of S-adenosylmethionine (SAM), the methyl donor exploited by both histone and DNA methyltransferases. Ascorbate and 2-oxogluatarate sustain ten-eleven translocation protein (TET) activity, a family of enzymes deputed to carry out the DNA demethylation cycle. Studies analyzing the effect of supplementation of these molecules are still preliminary and have mainly been conducted in vitro and in pre-clinical experiments promising interesting outcomes, especially on the viability of cancer cells. Here, recent pieces of evidence related to this novel epi-metabolic nutritional approach directed against colorectal and pancreatic cancer will be discussed. Most of these studies exploited GI cancer mouse models to investigate the therapeutic potential of the epi-metabolic approaches. These are pioneering studies, which underline epigenetic pathways as potential therapeutic targets of GI cancer. In this light, further investigation is needed to define the specific molecular players and develop appropriate therapeutic protocols for the transferability to clinics. The passage to the clinic is not automatic and needs specific evaluation.

The anti-neoplastic properties of curcumin are achieved by the inhibition of molecular pathways involving proliferation and inflammation and supporting the development of colorectal cancer. Guo et al. demonstrated a decrease in cancer incidence, colonic inflammation, and adenoma/adenocarcinoma appearance in a mouse model of colorectal cancer fed with a curcumin-supplemented diet [87]. This evidence was confirmed by Seiwert et al. by conducting similar in vivo experiments where colorectal cancer mouse models receiving curcumin in drinking water showed decreased signs of inflammation and smaller lesion sizes [88]. Moreover, preclinical studies analyzing the effect of the combination of curcumin with common chemotherapeutics for colorectal cancer, including 5-fluorouracil or oxaliplatin, showed beneficial effects able to reduce tumor volume and chemoresistance [89]. Between 2008 and 2020, six clinical trials (NCT02439385; NCT02724202; NCT03061591; NCT00927485; NCT00641147; NCT01490996) were activated to evaluate the potential of curcumin as an adjuvant for chemotherapy in colorectal cancer patients (Table 2). At the moment, it has been demonstrated that curcumin is safe and well-tolerated by humans [90,91,92], but its real therapeutic potential against colorectal cancer is still under investigation. Similar promising effects were documented in the prevention and treatment of pancreatic ductal adenocarcinoma (PDAC). Specifically, curcumin can re-sensitize chemoresistant pancreatic cancer cells to gemcitabine, the first line of chemotherapy used for pancreatic cancer, increasing its cellular uptake and potentiating its effectiveness [93]. It reduces desmoplasia, an intrinsic feature of PDAC, and inhibits tumor growth in combination with gemcitabine. Interestingly, two independent clinical studies showed that a combination of curcumin and gemcitabine increased the median survival rate of pancreatic cancer patients [90,91,92]. All of these pieces of evidence pointed out the promising potential of curcumin in colorectal and pancreatic cancer, two extremely aggressive GI cancers.

Likewise, sodium butyrate has shown anti-neoplastic effects. Specifically, it is able to suppress colorectal cancer cell proliferation and reduce the expression of stem markers, favoring differentiation and ultimately affecting colorectal cancer plasticity. Sodium butyrate exerts a metabolic effect increasing oxidative pathways and decreasing glycolytic metabolism in colorectal cancer cells [94,95,96]. Furthermore, the ability of sodium butyrate to activate both intrinsic and extrinsic apoptotic pathways in human pancreatic cancer cell lines has been demonstrated, showing its potential as a novel supplementation nutrient able to improve chemotherapy and immunotherapy in pancreatic cancer [97].

Physiological DNA methylation represents one of the mechanisms ensuring genomic stability, a driving force against tumorigenesis. Indeed, malignant cells show aberrant DNA methylation patterns in both colorectal adenoma and carcinoma [98].

As mentioned above, methionine and folate are approved nutritional supplements serving as precursors for the methyldonor SAM, which showed anti-tumoral effects in GI cancer [99,100,101]. It has been demonstrated that methyl donor supplementation has a significant inhibitory effect on the early phases of colorectal carcinogenesis, both in vivo and in vitro in a 3D-colorectal cancer cell model [101]. Moreover, a clinical study conducted on patients affected by resected colorectal cancer treated with a combination of chemotherapy and SAM showed that its supplementation decreased chemotherapy side effects, including liver toxicity, ultimately reducing chemotherapy course delays, discontinuations, and dose reductions [102]. Instead, for what concerns folate and methionine supplementation in pancreatic cancer, only a few studies at present have correlated their dietary intake with a reduced risk of its onset, whereas no study has been conducted, at present, to analyze their effect in the management of these oncological patients or in combination with gemcitabine [103].

Ascorbate supplementation is achieved by Vitamin C enrichment into the diet, an essential nutrient whose deficiency associates with several health problems due to its role in demethylation and oxidative stress reactions. However, its anti-neoplastic outcome is still controversial and mostly depends on both delivery and dosage [104]. Interestingly, TET activity, a family of enzymes deputed to the DNA demethylation cycle, is significantly reduced in several solid tumors [105,106,107]. It has been demonstrated that TET function alteration impaired the expression of different chemokines and the adaptative immune response suppressor, programmed death ligand 1 (PD-L1), explaining one of the mechanisms used by tumors to develop immunotherapy resistance [108]. In this light, TET activity levels might be exploited as a biomarker for immunotherapy response and need to be stimulated to improve its positive outcome. In a mouse model of colorectal cancer, Vitamin C supplementation at a high dosage is able to stimulate the immune anti-tumor activity and potentiate the immunotherapy response restoring chemokine expression levels and increasing tumor lymphocyte infiltration [108]. Similarly, in preclinical mouse models of colorectal and pancreatic cancer, high dosage Vitamin C supplementation impaired tumor growth, enhancing adaptative immune cell homing into the tumor and improving tthe therapeutic potential of immune checkpoint inhibitors [109]. Moreover, recent work from Eyres et al. indicates the role of TET in the switch of pancreatic adenocarcinoma from the squamous to the classical subtype, opening the avenue for therapeutic interventions able to restore a more favorable pancreatic cancer phenotype, which responds more to therapy and shows a longer survival rate [110]. In particular, it has been demonstrated that supplementation of Vitamin C combined with Metformin, a known anti-diabetic drug, is able to restore TET activity and consequently 5-hydroxymethylation levels, specifically in the promoter of genes involved in stimulating the differentiation phenotype of PDAC [110]. Recent literature on the anticancer therapeutic potential of Vitamin C put under the spotlight the DNA methylation cycle as a potential molecular GI cancer vulnerability. However, all these investigators stressed that only a high dosage of Vitamin C exerts the anti-neoplastic effect, an aspect to be carefully evaluated for use in clinics. These pieces of evidence indicated, for the first time, that epigenetics and metabolism cooperate to harness events determining the differentiation phenotype of specific tumors and to ultimately affect their therapeutic response and clinical outcome.

## 5. Conclusions

The link between GI cancer and diet has been established. Diet composition impinges on GI cancer onset, progression, and therapy response so much that different clinical studies are analyzing the effect of specific nutritional interventions and management on the chemotherapy tolerability and survival rate.

The present review summarized the clinical nutritional problems associated with GI cancer and the related clinical studies dealing with nutritional interventions in these oncological patients. Interestingly, all these studies highlight the importance of the early evaluation of GI cancer patients’ NIS and their nutritional status to adopt specific nutritional management. Indeed, nutritional interventions influence the quality of life of these patients and their tolerability to antineoplastic treatments. For these reasons, the figure of a dietician is essential to support these patients at the moment of diagnosis and during chemotherapy. The follow-up of the nutritional status is an overlooked aspect of GI cancer patients, which instead needs particular attention. In this light, the development of tools able to carefully evaluate the effect of nutritional interventions is an unmet need for GI cancer. What oncologists need to develop in the future is the design of patient-tailored nutritional management to place side-by-side conventional anti-neoplastic protocols.

Furthermore, the revision of recent literature has given an overview about the potential of nutritional interventions targeting the epi-metabolic link, whose alteration contributes to the onset, progression, and chemotherapy response of GI cancer. However, most of these studies were performed in GI cancer animal models. Therefore, further investigations are indispensable to establish their therapeutic potential and to define their transferability to clinics.

## Figures and Tables

**Table 1 nutrients-14-01542-t001:** Clinical trials for GI cancer patients concerning nutritional expedients.

NCT Number	Title	Status	Start	Country
NCT02066363	Study of parental nutrition to patients with GI cancer	completed	2014	Denmark
NCT03863236	A study of perioperative oral nutritional support for patients having surgery for colon cancer, Peri-Nutri	recruiting	2019	Finland
NCT04218253	Clinical application of nutrition support package before hepatectomy	recruiting	2019	China
NCT03930888	Nutritional support in patients undergoing surgical treatment of colorectal cancer	completed	2019	Czech Republic
NCT02681601	Nutrition support to improve outcomes in patients with unresectable pancreatic cancer	active	2016	USA
NCT04201730	Study on the efficacy and safety of enhanced recovery after surgery (ERAS) in GI cancer	completed	2019	China
NCT04190121	Perioperative nutritional support in Esophageal cancer patients	recruiting	2019	Greece
NCT01704664	Perioperative immunonutrition and phagocytic and bactericidal activity of blood platelets in gastric cancer patients	recruiting	2007	Poland
NCT00003851	Gemcitabine compared with pancreatic enzyme therapy plus a specialized diet (Gonzalez regimen) in treating patients who have stage II, stage III, or stage IV pancreatic cancer	terminated	1999	USA
NCT03949907	Early intravenous administration of nutritional support	recruiting	2020	Italy
NCT02788955	Protein recommendation to increase muscle	recruiting	2016	Canada
NCT01218841	Pure fish oil parental lipid emulsion in patients with GI cancer	completed	2005	Brazil
NCT04732442	Changes in inflammatory response after immunonutrition compared to standard nutrition in colorectal cancer tissue	completed	2017	Poland
NCT04607057	Supplemental parental nutrition during postgastrectomy in nutritionally at-risk patients	recruiting	2020	Korea
NCT05030090	Integrative nutrition care plan for a patient with liver and colorectal cancer	enrolling	2021	Taiwan
NCT01830907	Efficacy of preoperative nutritional support on the postoperative outcome in gastric patients at nutritional risk by NRS-2002	completed	2012	China
NCT04513418	Effects of preoperative enteral immunonutrition for esophageal cancer patients given neoadjuvant chemoradiotherapy	recruiting	2020	China
NCT02626195	Preoperative nutritional support in malnutritional cancer patients	completed	2013	Korea
NCT01276795	Whey protein-based enteral nutrition support to improve the protein economy in surgical patients	completed	2010	Canada
NCT03150615	Enteral nutrition after pancreaticoduodenectomy	completed	2016	China
NCT01222208	Oral versus parental nutrition support to improve protein balance in colorectal surgical patients	completed	2011	Canada
NCT00919659	Parental nutrition support for patients with pancreatic cancer	completed	2002	Germany
NCT01870817	Home jejunostomy feeding following esophagectomy/gastrectomy	completed	2012	UK
NCT04567459	The effect of nutrition for colorectal cancer patients receiving chemotherapy; randomized controlled study	recruiting	2021	Taiwan
NCT04188990	Cost effectiveness of an intervention in hospitalized in patients with disease-related malnutrition	recruiting	2020	Spain
NCT04109495	Usefulness of a smartphone application for improving the nutritional status of pancreatic cancer patients	completed	2017	Korea
NCT02155140	Enteral feeding in discharged patients	terminated	2011	UK
NCT03550482	Oncoxin and quality of life in cancer patients	completed	2017	Russia
NCT04131426	Evaluating the combined intervention of nutritional supplementation (Remune) and exercise in patients with cancer cachexia	recruiting	2020	USA
NCT04597151	Diet education program for stage I–IV colorectal cancer survivors	recruiting	2020	USA
NCT03475966	Improving outcomes in cancer patients with a nutritional and physical conditioning prehabilitation program	recruiting	2017	Canada

**Table 2 nutrients-14-01542-t002:** Clinical trials evaluating curcumin as an adjuvant for chemotherapy in colorectal and pancreatic cancer.

NCT Number	Title	Status	Start	Country
NCT02439385	Avastin/FOLFIRI in Combination with Curcumin in Colorectal Cancer Patients with Unresectable Metastasis	Completed	2015	Korea
NCT02724202	Curcumin in Combination with 5FU for Colon Cancer	Active	2016	USA
NCT03061591	Turmeric Supplementation on Polyp Number and Size in Patients with Familial Adenomatous Polyposis.	Active	2017	Israel
NCT00927485	Use of Curcumin for Treatment of Intestinal Adenomas in Familial Adenomatous Polyposis (FAP)	Completed	2009	Puerto Rico
NCT00641147	Curcumin in Treating Patients with Familial Adenomatous Polyposis	Completed	2008	Puerto Rico
NCT01490996	Combining Curcumin with FOLFOX Chemotherapy in Patients with Inoperable Colorectal Cancer (CUFOX)	Completed	2011	UK

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
