# Peer review of "Gastrointestinal Cancer Patient Nutritional Management: From Specific Needs to Novel Epigenetic Dietary Approaches"

_nutrients, 2022, doi:10.3390/nu14081542_

Round 1
Reviewer 1 Report
Review Nutrients 2022. Gastrointestinal Cancer patient and nutritional management.
Interesting paper, highlighting the nutritional challenges in patients with GI cancer and possible ways of using nutrition. High quality studies in nutritional management is still needed. This gives an idea of some of the previous evidence, primarily from studies on mice, and direction for possible further studies.
Be clear about the aim; discussion of previous studies on nutritional therapy with effect on epi-genetic and metabolic status in patients with GI cancer?
The grammar and tense needs to be checked upon before submission.
Abstract
Line 28-30; this sentence has to be re-written to be clear to the reader what the main aim for the study is. Mix of past and present tense; Recently…., are.
- Introduction
Consider to mention Nutrition impact symptoms (NIS) when describing the reasons for malnutrition in patients with cancer, which is multifactorial (line 68 -69).
Line 35-38 unclear; re-write.
Line 46 GI-cancers does as well include the small intestine.
Line 58-62 Revision, sentence doesn’t make sense. EX. Studies has addressed the effects of various nutritional methods in responsiveness to prevent disease progression and/ or reduce side effects to chemotherapy.
- Diet intervention in cancer therapy
Line 105-106; revise; not clear. Evaluation of recent clinical trials on nutrition in patients with GI cancer has mainly included studies on …………………
Line 139-141; last sentence very long, delete the last phrase. Or …in these patients, who are in significant risk of malnutrition.
- Diet as intervention
Line 157 – 160 definition of sarcopenic obesity; loss of muscle mass or decline in muscle function combined with obesity.
Line 190 -194; very long sentence
- Epi-metabolic diet approaches.…..
Important to stress that studies in mice models cannot translate directly to humans. It may be that some of the evidence referred to in this article is from studies in mice models. It’s not transferrable to humans, it will have to be verified in human studies before the evidence is strong enough to prove anything.
Conclusion
include a discussion of the studies referred to. How good is the evidence, does it forward any new knowledge that may me usefull in future treatment regimes. There is an urgent need of new studies in the field, in combination with personalized dosage in may be one of the keys to change the epi-genetics for better prognosis.
Author Response
The authors would like to thank the Editor and the Reviewers for their useful comments on our work.
Review Nutrients 2022. Gastrointestinal Cancer patient and nutritional management.
Interesting paper, highlighting the nutritional challenges in patients with GI cancer and possible ways of using nutrition. High quality studies in nutritional management is still needed. This gives an idea of some of the previous evidence, primarily from studies on mice, and direction for possible further studies.
Q1. Be clear about the aim; discussion of previous studies on nutritional therapy with effect on epi-genetic and metabolic status in patients with GI cancer?
A1. We would like to thank the Reviewer for His/Her valuable comment. The revised manuscript has been edited according to the Reviewer’s suggestions.
Q2. The grammar and tense needs to be checked upon before submission.
A2. The revised manuscript has been checked according to the Reviewer’s suggestions.
Q3. Abstract
Line 28-30; this sentence has to be re-written to be clear to the reader what the main aim for the study is. Mix of past and present tense; Recently…., are.
A3. The sentence has been rewritten accordingly.
Q4. Introduction
Consider to mention Nutrition impact symptoms (NIS) when describing the reasons for malnutrition in patients with cancer, which is multifactorial (line 68 -69)
Line 35-38 unclear; re-write.
Line 46 GI-cancers does as well include the small intestine.
Line 58-62 Revision, sentence doesn’t make sense. EX. Studies has addressed the effects of various nutritional methods in responsiveness to prevent disease progression and/ or reduce side effects to chemotherapy.
A4. The Introduction section of the revised manuscript has been modified according to the Reviewer’s recommendations.
Q5. Diet intervention in cancer therapy
Line 105-106; revise; not clear. Evaluation of recent clinical trials on nutrition in patients with GI cancer has mainly included studies on …………………
Line 139-141; last sentence very long, delete the last phrase. Or …in these patients, who are in significant risk of malnutrition.
A5. The paragraph entitled Diet interventions in cancer therapy of the revised manuscript has been edited
according to the Reviewer’s comments.
Q6. Diet as intervention
Line 157 – 160 definition of sarcopenic obesity; loss of muscle mass or decline in muscle function combined with obesity.
Line 190 -194; very long sentence
A6. The paragraph entitled Diet as intervention of the revised manuscript has been modified according to the Reviewer’s suggestions.
Q7. Epi-metabolic diet approaches.…..
Important to stress that studies in mice models cannot translate directly to humans. It may be that some of the evidence referred to in this article is from studies in mice models. It’s not transferrable to humans, it will have to be verified in human studies before the evidence is strong enough to prove anything.
A7. The paragraph entitled Epi-metabolic diet approaches of the revised manuscript has been modified stressing that discussed studies were performed mainly in mouse models of GI cancer and that further investigations about transferability to clinics need be specifically assessed.
Q8. Conclusion
include a discussion of the studies referred to. How good is the evidence, does it forward any new knowledge that may me usefull in future treatment regimes. There is an urgent need of new studies in the field, in combination with personalized dosage in may be one of the keys to change the epi-genetics for better prognosis.
A8. The paragraph related to the Conclusion section has been edited summarizing what is the state of the art related to nutritional interventions in GI cancer. The need for further studies on the topic has been stressed, discussing which are the most urgent aspects to face in the future. The epi-metabolic approaches have been contextualized as evidence mainly obtained in pre-clinical studies with promising clinical outcomes to be verified. Specifically, a point on the clinical transferability of these results has been stated.
Reviewer 2 Report
Comprehensive review, further immanent changes/additions are needed to improve the quality of the manuscript:
- Linguistic revision through an english native speaker
- Cancer cachexia in relation to fat and muscle metabolism/inflammation and immunity - link to chemotherapy
- Specific sections for curable vs. advanced/metastasized cancer
- Concluding points/graph as basis/outline for further research
- Minor points:
- Quote literal citations (line 75-87, Fearon et al.)
- Explain TET (abbreviation, line 392)
Author Response
The authors would like to thank the Editor and the Reviewers for their useful comments on our work.
Comprehensive review, further immanent changes/additions are needed to improve the quality of the manuscript:
We would like to thank the Reviewer for His/Her valuable comment.
Q1. Linguistic revision through an english native speaker
A1. Grammar and tense of the present manuscript have been checked according Reviewer’s suggestions.
Q2. Cancer cachexia in relation to fat and muscle metabolism/inflammation and immunity - link to chemotherapy
A2. In the revised version of the present manuscript, cancer cachexia has been contextualized in relation to fat, inflammation, and muscle function. The following references have been added to discuss this point raised by the reviewer:
- Fearon, K.; Arends, J.; Baracos, V. Understanding the mechanisms and treatment options in cancer cachexia. Nat Rev Clin Oncol 2013, 10, 90-99, doi:10.1038/nrclinonc.2012.209.
- Collins, P.; Bing, C.; McCulloch, P.; Williams, G. Muscle UCP-3 mRNA levels are elevated in weight loss associated with gastrointestinal adenocarcinoma in humans. Br J Cancer 2002, 86, 372-375, doi:10.1038/sj.bjc.6600074.
- Kandarian, S.C.; Nosacka, R.L.; Delitto, A.E.; Judge, A.R.; Judge, S.M.; Ganey, J.D.; Moreira, J.D.; Jackman, R.W. Tumour-derived leukaemia inhibitory factor is a major driver of cancer cachexia and morbidity in C26 tumour-bearing mice. J Cachexia Sarcopenia Muscle 2018, 9, 1109-1120, doi:10.1002/jcsm.12346.
Although different pieces of literature deal with this specific topic, we decide to limit discussion of this aspect for the sake of not straying from the objectives of the present review.
Q3. Specific sections for curable vs. advanced/metastasized cancer
A3. We agree with the reviewer that this is an important aspect to consider, however, it could be the subject of a dedicated review. Indeed, “Clinical Implications of Malnutrition in the Management of Patients with Pancreatic Cancer: Introducing the Concept of the Nutritional Oncology Board.” Rovesti G, Valoriani F, Rimini M, Bardasi C, Ballarin R, Di Benedetto F, Menozzi R, Dominici M, Spallanzani A. Nutrients. 2021 Oct 7;13(10):3522 is a recently comprehensive published review on the topic.
Q4. Concluding points/graph as basis/outline for further research
A4. The paragraph related to the Conclusion paragraph has been edited summarizing what is the state of the art related to nutritional interventions in GI cancer. The need for further studies on the topic has been stressed, discussing which are the most urgent aspects to face in the future. The epi-metabolic approaches have been contextualized as evidence mainly obtained in pre-clinical studies with promising clinical outcomes to be verified. Specifically, a point on the clinical transferability of these results has been stated.
Minor points:
Q5. Quote literal citations (line 75-87, Fearon et al.)
A5. The present citation has been quoted at the end of the sentence.
Q6. Explain TET (abbreviation, line 392)
A6. The abbreviation TET has been explained according to the Reviewer’s suggestions.
Round 2
Reviewer 1 Report
Thank you for revisions of the manuscript.
The revisions have made the paper easy to easy, as well as the conclusions are now clear and in line with the aim.
I don’t have any further suggestions for revision, and do find the manuscript ready for publication.